# Shape Memory Polymers as Smart Materials: A Review

**DOI:** 10.3390/polym14173511

**Published:** 2022-08-26

**Authors:** Tarek Dayyoub, Aleksey V. Maksimkin, Olga V. Filippova, Victor V. Tcherdyntsev, Dmitry V. Telyshev

**Affiliations:** 1Institute for Bionic Technologies and Engineering, I.M. Sechenov First Moscow State Medical University (Sechenov University), Bolshaya Pirogovskaya Street 2-4, 119991 Moscow, Russia; 2Department of Physical Chemistry, National University of Science and Technology “MISIS”, 119049 Moscow, Russia; 3Institute of Biomedical Systems, National Research University of Electronic Technology, Zelenograd, 124498 Moscow, Russia

**Keywords:** activation mechanism, amorphous polymer, artificial muscles, electrical conductivity, one-way shape memory effect, semi-crystalline polymer, shape memory polymer, smart material, thermal conductivity, two-way shape memory

## Abstract

Polymer smart materials are a broad class of polymeric materials that can change their shapes, mechanical responses, light transmissions, controlled releases, and other functional properties under external stimuli. A good understanding of the aspects controlling various types of shape memory phenomena in shape memory polymers (SMPs), such as polymer structure, stimulus effect and many others, is not only important for the preparation of new SMPs with improved performance, but is also useful for the optimization of the current ones to expand their application field. In the present era, simple understanding of the activation mechanisms, the polymer structure, the effect of the modification of the polymer structure on the activation process using fillers or solvents to develop new reliable SMPs with improved properties, long lifetime, fast response, and the ability to apply them under hard conditions in any environment, is considered to be an important topic. Moreover, good understanding of the activation mechanism of the two-way shape memory effect in SMPs for semi-crystalline polymers and liquid crystalline elastomers is the main key required for future investigations. In this article, the principles of the three basic types of external stimuli (heat, chemicals, light) and their key parameters that affect the efficiency of the SMPs are reviewed in addition to several prospective applications.

## 1. Introduction

Shape memory polymers (SMPs) are smart materials that can be deformed and fixed into a temporary shape and can recover their permanent shape after the release of the external stimulus [1,2]. Because of their interesting properties, such as excellent structural versatility, lightweight, low cost, easy processing, high elastic strain (more than 200%), biocompatibility, and biodegradability, they attract much attention in the industrial, aerospace, textile, and medical fields [3,4,5,6,7,8]. A lot of external stimuli can be applied on the SMPs (Figure 1), such as heat [9,10,11], light [12,13], electricity [14,15,16,17], magnetic fields [18,19], chemical stimulus (pH changes) [20,21], humidity [22,23,24], etc.

Recently, many reviews about SMPs have discussed the aspects of SMPs, such as activation mechanisms [9,25], required molecular structure and changes through the activation of the shape memory effect [26,27,28], their applications, such as aerospace engineering [29,30,31,32], sensors and actuators [33,34,35,36], textile engineering [37,38,39,40,41], artificial muscles [42,43], packaging [44,45], etc., Figure 2. However, a simple and understandable clarification of the structural concepts and activation mechanisms is still missing.

Since SMPs can retain more than two of their temporary shapes during the shape memory effect (SME) process (two or more responses), they are considered convenient candidates for smart systems, which control different conditions and require multiple external stimuli [46]. However, there are some limitations, which obstruct the application of SMPs, such as low thermal and electrical conductivity, weak mechanical properties, and inertness to electromagnetic stimuli in comparison to well-researched shape memory ceramics and metallic alloys [47]. In the last two decades, comprehensive investigations have aimed at developing new polymers with improved SME for better performance [48,49,50,51]. In other words, modification of polymer structure using different processing methods and adding different fillers to improve the SMP properties, such as mechanical properties, electrical and thermal conductivity, etc., are considered the main direction of investigations and research all over the world to develop shape memory polymer composites (SMPCs).

Currently, the activation of SME in SMPs can essentially be carried out using three basic types of external stimuli, which are heat (thermo-responsive SMPs); chemicals (chemo-responsive SMPs); light (photo-responsive and photo-thermal responsive SMPs). Thus, this review will focus on the simple explanation of the main principles of SME activation by using the most familiar external stimuli and their latest applications in medical, aerospace and textile applications, Figure 3.

## 2. Activation Mechanisms of Shape Memory Polymers

In general, the shape changes under the effect of external stimuli, especially heat, electric field, and light, are related to the molecular architecture of the SMP, which consists of netpoints and switching domains [25]. As seen in Figure 4, netpoints can have a physical nature, such as entanglement coupling, crystalline phase, copolymers, or a chemical nature, such as covalent bonds, and the polymer chains among the netpoints are called switching domains. In the shape memory effect, the memorization of the original permanent shape of the SMPs is related to the netpoints that are responsible for permanent shape, whereas the reversible shape change of the SMPS is related to the switching domains that are responsible for temporary shape [9,52]. It should be noted that the flexibility of polymer chains in the switching domains is related to transition temperature (T_trans_), which will be equal to the glass transition temperature (T_g_) for amorphous polymers, or equal to the melt temperature (T_m_) for semi-crystalline polymers [52]. So, by heating SMPs higher than T_trans_, the polymer chains of the switching domains will have high flexibility and they can be deformed to obtain large strain at low stresses. After that, the polymer chains will be immobile, preserving the temporary shape when the polymer is cooled to a temperature lower than T_trans_, Figure 4. To return to the permanent shape, the polymer must be heated again to a temperature higher than T_trans_, at which the switching domains will acquire mobility and take a position corresponding to the permanent shape [9]. 

Depending on their response, shape memory polymers are divided into two groups. The first one is one-way SME, which is considered irreversible because of the polymer fixation in its initial shape that has a specific structure without the ability to repeat SME, since it requires reprograming the procedure to reactivate it, Figure 5. In other words, SME cannot be repeated by simply reversing the stimulus [25,53]. In contrast, the second one, two-way SME, is reversible with spontaneous shape change of materials [53,54,55] since it responds completely to the external stimuli by shifting between two separated shapes in a reversible way without the need for an additional reprogramming process, Figure 6a. In other words, when the SMPs with two-way SME are programmed and have learned specific behavior, it is possible to modify their shape in a reversible way between two different ones without applying stress or load, but only by using external stimuli such as heat. This type of SMP is considered a convenient candidate for soft actuators and sensors, artificial muscles, and soft robotics. According to the number of the temporary shapes that the SMPs can remember, they can be divided into dual-SMPs, in which the polymer can remember the temporary shape only, and multiple-SMPs, in which the polymer can remember two or more temporary shapes and restore the initial shape step-by-step [25,26,53,54,55].

At the structure level, for two-way shape memory polymers, the activation mechanism of SME is considered complicated and dependent on the polymer type. In general, there are four types of polymers: semi-crystalline polymers, liquid crystalline elastomers (LCEs), multi-layered polymer composites, and interpenetrating polymers that consist of elastomeric and crystalline networks, Figure 6. 

For semi-crystalline polymers, the SMP consists of an internal skeleton that is made up of netpoints, crystallites (lamellae), and switching domains (polymer macromolecules in amorphous phase), which form the entire framework. In the isotropic state, the lamellae will be randomly located in the polymer structure, Figure 6b. In the programming stage, by heating the polymer to a temperature higher than T_trans_ (higher than T_m_), the crystallites will melt, then by cooling the polymer under constant tension, new oriented crystallites will be formed with a bigger size in comparison with the polymer crystallites in the isotropic state, which must have a wide range of melting temperature (Figure 6b). These crystallites will be oriented along the stretching direction during the cooling process under a fixed strain, which in its turn leads to a decrease in the system entropy. This process is called crystallization-induced elongation [26]. So, in the two-way SME and by heating the polymer to its T_trans_ (T ≤ T_m_), a part of the polymer macromolecules in the crystalline phase will melt, after that they will change into the amorphous phase, leading to a shrinkage (contraction) of the polymer as a result of the melting-induced contraction process, which in its turn will lead to an increase in the system entropy; then by cooling the polymer, the melted macromolecules will be crystallized again under conditions of self-nucleation. The self-nucleation in the crystallization process is caused by the remaining part of the crystalline phase and this effect leads to the crystallization of the oriented crystallites in the stretching direction of the programming stage. Self-nucleation ensures the existence of the “memory” of the previous crystalline state [27] and the semi-crystalline polymer elongates again towards the crystallization direction. In the case of two-way SME and zero-stress conditions, the polymer internal skeleton is eligible for regulating and retaining the alignment of switching domains during multi- heating and cooling cycles at temperatures range of T ≤ T_trans_, and it is sufficiently elastic to prevent the recovery of the initial polymer shape [28]. This means that during heating, switching domains can contract in the direction of its alignment, causing shrinkage of the skeleton frame, while their macromolecules can expand in the direction of its alignment, causing an extension of the skeleton frame during cooling [56]. Moreover, these structure changes lead to a stress formation in the polymer volume, therefore, to conserve the two-way SME, the formed stresses must be equivalent to the ones caused by entropy elastic recovery [57].

Liquid crystalline elastomers (LCEs) consist of elastomeric chains as a backbone and cross-linked sidechains. They are produced based on the crosslinking reaction of reactive-mesogenic monomers. The activation mechanism of SME in this polymer type is based on the loss and restoration of the molecular groups’ alignment (the orientation of mesogens) because of the mesomorphic/isotropic phase transitions after applying external stimuli, Figure 6c [58,59]. Anyhow, under external stimulus, poly-domains, the randomly oriented liquid crystalline domains, can be reoriented in a particular direction—mono-domains [60,61]. As for semi-crystalline polymers, in the programming stage by heating LCEs to a temperature higher than T_trans_ and under application of a constant tension, the mesogens will be oriented and aligned in the direction of the stress, which leads to a crystallization-induced elongation process during the cooling process under a fixed strain. This means that the un-melted domains will be responsible for the geometry shift of the network during the cooling process. By heating the LCEs to their T_trans_, the melted domains decrease their anisotropy due to the increase in the elasticity modulus of the isotropic network, which leads to the contraction of the material because of a melting-induced contraction process. 

Multi-layered polymer composites with two-way SME, generally, are prepared as non-covalent linkage composites by bonding two or multi-polymeric layers using adhesives. These polymer layers can be made of one-way SMPs or one-way SMP/elastomeric polymer composites, Figure 6d [62,63]. For the preparation of one-way SMP/elastomeric polymer composites, the one-way SMP will be deformed at a temperature higher than its T_trans_ and then cooled down to fix its deformed shape. After that, the elastomer layer will be adhesively fixed onto the deformed SMP layer. So, by heating the prepared composite to activate the two-way SME in these composites, the deformed SMP layer will recover its initial shape, which leads to a tension applied to the elastomer layer causing bending of the composite. Whereas by cooling the composite, the SMP layer will lose its recovery tension, so that the initial shape of the multi-layered composite is recovered. However, the expansion of the elastomer layer during the heating process and weak interfacial bonding among composite layers are considered the limitations for applying this type of two-way SMP [64].

In contrast, interpenetrating polymers, the modern type of two-way SMP, do not require a programming stage or external tension, therefore, they are formed by the combination of elastomeric and crystalline networks using a mixing method; only the physical bonding among them used polymer networks, Figure 6e [65,66,67,68]. The activation mechanism of SME here is dependent on switch–spring composition modeling, in a way that the elastomeric network will behave as a spring and the crystalline network will act as a switch, Figure 6e. So, by heating the crystals to T_trans_ of the crystalline network, they will melt, and the chains will have a high flexibility (the switch part will be opened), which will lead to the shrinkage of the composite and simultaneously to the compression of the elastomer (spring compression). However, by cooling the composite to the switching temperature of the crystalline network, the polymer macromolecules will be crystallized in the direction of the spring force due to the elastic recovery of the elastomer. It should be noted that the T_trans_ values for two-way SMPs based on interpenetrating polymers are lower than the ones of the semi-crystalline polymers and LCEs, and this type of SMP does not lose its two-way SME at high temperature. 

Generally, materials prefer stability, which means that material behavior is driven by reaching the lowest possible energy state. In the programming stage of SMPs, a high energy will be transformed within the material, which makes it metastable; and the material will tend to release this energy, which is considered the driving force of actuation [69,70]. Physical and chemical cross-linking points, previously mentioned, are responsible for the permanent shape of SMPs, by preventing the polymer chains from slipping, which leads to a decrease in the entropic energy stored within the polymer network and the prevention of shape recovery [69,70]. Moreover, during the programming process, a metastable configuration, within the SMP, will be formed, and a reversible fixing process will be required to utilize the imparted energy, commonly by using thermal transitions. Therefore, shape recovery of SMPs will occur because of the disruption of this reversible process [71]. 

SMPs can respond to some external stimuli mentioned above in Figure 1 and heat is the most common stimulus used to activate SME in SMPs. In the next items, the activation mechanisms for SMPs using common external stimuli are reviewed.

### 2.1. Activation by Heating 

Activation of SMPs using heat is considered the most common method among other external stimuli. The thermal activation mechanism can be divided into direct and indirect depending on the heating methods, such as electro-resistive heating and inductive heating.

#### 2.1.1. SMP Activation by Direct Heating 

In the direct heating method, absorption or excretion of the thermal energy through the matrix of SMPs are considered controlling factors. Since heat will transfer through the polymer matrix leading to changes in the polymer structure and the SME activation, by heating the polymer higher than its T_trans_ as previously mentioned. Depending on the type of the polymer structure, T_trans_ can be the temperature of the glass transition (T_g_) for the amorphous polymers or the melting temperature (T_m_) for semi-crystalline polymers and copolymers, and covalently cross-linked polymers. 

For amorphous polymers, heating the polymer to its T_g_ gives the possibility to control its chain mobility, which means the ability to reduce the entropy by fixing its chain network. Whereas heating the amorphous polymer to a temperature higher than its T_g_ leads to unconstrained shape recovery. This is related to the tendency of the amorphous polymer network to increase the entropy when the crosslinking points return to their original positions after heating [70,72,73]. 

Based on the previously defined responsibilities of netpoints and switching domains, by heating the polymer to its melting temperature, the crystalline phase will be softened leading to the SMPs initial shape recovery for the semi-crystalline and covalently cross-linked polymers [70,71,72,73,74].

In the SME process, after heating the SMP to a temperature higher than its T_trans_, the polymer chains can be deformed to obtain a temporary shape. In this stage, called the entropic elasticity, polymer macromolecules in the switching domains will be oriented because of the elastic deformation that leads to a decrease in the specific entropy of the SMP. Since the SME in the SMPs is driven by the entropic elasticity of the switching domains, the polymer macromolecules in the amorphous phase of the SMPs show a random coil conformation that refers to the state of highest entropy according to the Boltzmann Equation (1) [9,75]. By releasing the applied stress when the polymer temperature is higher than T_trans_, the polymer macromolecules in the polymer amorphous phase will restore their random coil conformation in addition to their high entropy, which leads to recovery of the original shape of the polymer. Thus, this immediate recovery of the polymer shape can be obstructed by cooling the polymer to a temperature lower than its T_trans_, so that the macromolecules of the switching domain will be frozen (solidification) [70,71,72,73,74].
S = κ lnW(1)
where, S—entropy, κ—Boltzmann constant, W–represents the possibility of a conformation of the polymer chain. 

#### 2.1.2. SMPs Activation by Indirect Heating

Generally, activation of the stretch/compression process (SME) using direct heating is carried out by using hot and cold medium (air, water, or liquid). However, taking into consideration the low thermal conductivity of SMPs, the main problem limiting SMP activation by direct heating is when hot/cold medium is applied to the SMPs, the heating/cooling rate on the surface and inside the polymer can differ significantly, which leads to an uneven temperature distribution over the volume of the polymer, which in turn is accompanied by overheating of the polymer surface and the beginning of surface actuation before the compression of the inner part of the SMPs. This will lead to an increase in the time of polymer exposure to a hot/cold medium until the polymer volume has fully activated, which means a slow actuation response. These disadvantages mean that using this activation method is not suitable for all types of applications. Therefore, using indirect heating, such as electro-resistive heating and inductive heating, can be considered a potential method to activate the SME in SMPs. The main aim of the SMPCs preparation is to improve the electrical and thermal conductivity of SMPs without a perceptible decrease in their mechanical properties, by adding fillers into the polymer matrix, such as carbon fillers and metals. These have high thermal and electrical conductivity and can transform the applied stimulus (electric, magnetic, etc.) on the SMPCs into thermal energy which is the main factor in obtaining a good dispersion of the added fillers [76,77]. 

Electro-induced shape-memory polymeric actuators are a kind of polymer with SME that can be activated by electrical voltage to initiate the actuation process and the deformation of the polymer composite via Joule heating through a fully actuator volume at the same time [78,79]. Therefore, to obtain electrically conductive polymer composites, it is necessary to add electrically conductive fillers, such as carbon nanotubes (CNTs), graphene, carbon black, and metallic fillers, to the polymer matrix to prepare a percolated electric conductive polymeric composite that leads to a uniform heating through the polymer volume when an electrical voltage is applied. When a continuous network of filler is formed, the polymer material acquires electrically conductive properties. In the case when observing a sudden decrease in the electrical resistance of the material, the minimum filler concentration is called the percolation threshold. The percolation threshold is a critical parameter as it determines the final cost of the composite. The percolation threshold depends on the filler type and shape, and on the method of introducing the filler into the polymer matrix, which determines its spatial distribution in the polymer. Modern electrically conductive polymer composite materials can have low values of electrical conductivity, such as in the following examples: for composites based on polyethylene and 24% copper, the electrical conductivity is 0.11 S cm^−1^ [80], for polypropylene filled with 80% tin-lead alloy the electrical conductivity is 90 S cm^−1^ [81] and for polypropylene filled with 40 wt% carbon black the electrical conductivity is 10^−5^ S cm^−1^ [82].

In the reference [83], the authors prepared a nanocomposite, which consists of poly (lactic acid)/epoxidized soybean oil/carbon nanotubes (PLA/ESO/CNTs) using solution mixing. Since the CNTs addition aimed at enhancing the electrical conductivity of the composite, the authors showed that the ESO addition into their composite led to the enhancement of the composite flexibility to be used as an electroactive shape recovery material. Their prepared composite had a conductivity of 10^−4^ S m^−1^, which under a constant voltage of 60 V shape would be recovered within 12 s. In the reference [84], electro-thermal actuator was prepared by adding super aligned carbon-nanotube sheets (randomly oriented CNTs) into poly(dimethyl siloxane) (PDMS) layers. The authors showed that their composite could achieve a bending angle over 540° under a 12 V driving voltage. This deformation needed about 20 s and the prepared composition could rapidly return to its initial state because of its fast heat dissipation when the applied voltage was switched off (Figure 7).

In the reference [85], 3D printed conductive polylactic acid (CPLA) was used as a self-pneumatic actuator (SPA). Since the CPLA was stiff, the SPA would not be able to bend even when the pressure input was turned on. In order to activate Joule heating, silver wires were used and then the CPLA with the silver wires was encapsulated through a silicone rubber bath to allow adhesion with the SPA. The authors showed that the prepared material was actuated by 12 V, and it showed 98.6% reduction of Young’s modulus by heating up to 80 °C, which was completely recovered after cooling the material to the initial temperature. In the reference [86], a segregated structure of CNTs (S-CNT) was added to a semi-crystalline polymer of poly(ethylene-co-octene) (POE) to prepare multi-responsive reversible shape-memory polymers (RSMPs). The authors reported that the prepared composite had low modulus but high conductivity (up to 0.046 S cm^−1^ with a CNT content of 2 vol%) with rapid response at low driving voltage (≤36 V). As it can be seen in Figure 8, a mechanical gripper based on prepared POE/S-CNT composite could open its fingers over 18 s under a voltage of 36 V and close within 168 s when the voltage was turned off, while 186 s was needed for the entire cycle.

However, the main problem of using carbon materials as fillers in SMPs is the formation of aggregation in the polymer matrix, which is related to the Van der Waals forces of the carbon fillers [87]. The formation of the filler aggregation leads to uneven dispersion of filler, which in turn leads to loss of the percolated network inside the polymer matrix and impairment of the SMPC mechanical properties. Moreover, a high electrical field is required to activate the SME of SMPCs, which can cause undesirable electric shocks, and the production of monopolar actuation because of the associated electrostriction effect [88,89].

Another promising method for thermal activation of SMPs is the use of thermal inductive heating. Inductive heating is the process of material heating using electrically conductive materials due to eddy currents (Foucault’s currents) induced using an alternating electromagnetic field. In this case, the induced eddy currents are completely closed inside the heated material. The main factor in inductive heating is the electrical resistivity of the material, which means that it should have a high electrical resistivity in order to be heated faster and better compared to the one with low electrical resistivity. In this case, there is no need for a direct contact between the SMPs and the inductor coil.

An installation schematic diagram for inductive heating is shown in Figure 9a. The inductor itself, into which the heated electrically conductive material is placed, is made in the form of a copper coil, through which an alternating current flows at a certain frequency, thereby generating a high-frequency electromagnetic field. An alternating electromagnetic field induces eddy currents in an electrically conductive material, Figure 9b, which leads to the action of heating the material. One of the key parameters of the flowing electric current through the coil of the inductor is its frequency. The occurrence and magnitude of eddy currents induced in the heated material are directly related to the frequency of the inductor current, which means that eddy currents are also variable and have the same frequency as the current flowing through the coils of the inductor, while the density of the induced eddy currents is proportional to the frequency of the inductor current. In modern inductors, currents with a frequency from tens of Hz to several MHz can be generated.

There are several works in which inductive heating had been used to thermally activate SMPs. In the reference [16], polycaprolactone was used as a shape memory polymer and Fe_3_O_4_ nanoparticles (11 nm) were used as micro-particles heated under the action of an electromagnetic field. The mass fraction of Fe_3_O_4_ particles was 40 wt%. Figure 10 shows a schematic diagram of the installation and photographs showing the process of returning to the original (programmed) shape of the material when heated with an inductor. The heating of the material was carried out at temperature of 46 °C and the heating time was 12 s. 

In the reference [90], the authors prepared Fe_3_O_4_/poly (ε-caprolactone)-polyurethane (PCLU) shape memory nanocomposites using an in-situ polymerization method. They showed that the prepared nanocomposites started to recover their permanent shape near 40 °C in 60 s by applying an alternating magnetic field of 45 kHz with a recovery rate of 97%, Figure 11.

In the reference [91] composites of carboxylic styrene butadiene rubber (XSBR)/ferriferrous oxide (Fe_3_O_4_)/zinc dimethacrylate (ZDMA) with dual response SME were prepared, Figure 12. The authors demonstrated that the ZDMA addition led to the decrease of Fe_3_O_4_ aggregation in the polymer matrix and the T_g_ of the prepared composites could be regulated from 20.5 to 32.3 °C by changing the dosage of ZDMA. Moreover, the authors claimed that the tensile strength of the prepared composite improved from 15.12 MPa for neat XSBR to 30.26 MPa for XSBR/70 phr Fe_3_O_4_/10 phr ZDMA, and the recovery rate achieved up to 100% using both direct heating and magnetic stimulus. 

In general, the advantages of the thermally activated SMPs using inductive heating are the high heating efficiency (>90%), the non-contact heating; the possibility of rapid heating of the SMPCs in a few seconds, and the ability to control the heating rate and heating temperature by changing the frequency of the electromagnetic field. On the other hand, the disadvantages of these polymers are the generation of a strong electromagnetic field, which can interfere with nearby electrical equipment, and the need for large-size materials. In other words, inductive heating for small-size SMPC materials will decrease the eddy current circuit, which will reduce the heating efficiency.

### 2.2. Chemical Activation by Solvents

The main role of the solvent used for the activation of SMPs is the decrease of the T_g_ of the polymer. This means by using a solvent (water or organic solvents), its molecules will be absorbed into the polymer matrix, leading to an interruption of the secondary bonding between the polymer macromolecules, which in turn will lead to polymer swelling, i.e., the activation mechanism will be related to the swelling/shrinkage process [92,93,94]. Thus, the solvent will behave as a plasticizer, leading to enhancement of the chain mobility of the polymer by increasing the configurational entropy of the system, which in turn will lead to a reduction in the relaxation time and the glass transition temperature of the polymer [95].

#### 2.2.1. Chemical Activation by Water

As mentioned above, water has a plasticizing effect on the SMP, i.e., the water molecules can penetrate the amorphous phase of the polymer, leading to improvement of the polymer chain mobility, which will lead to a decrease in the T_g_ of the polymer [96]. Polyvinyl alcohol [97], composites based on polyurethane [98] and polyethylene glycol [99] are considered typical shape memory polymers that have water response. 

Zhou et al. [97] used poly(vinyl alcohol) (PVA), cross-linked by ureido–pyrimidinone (UPy) dimers to prepare shape memory material that presented excellent thermo- and water-induced shape memory behavior with a recovery ratio up to 99%. The authors showed that the absorbed water led to a significant decrease in the storage modulus values, as a result of a possible shape recovery. Moreover, they claimed that the shape recovery process of their prepared material occurred in water at 16 °C, and the material recovered its permanent shape within 30 min after water absorption of about 50%. 

In the reference [98], the authors prepared bio-based material based on polyhydroxyalkanoate (PHA), polyurethane (PHP), and polyethylene glycol (PEG). They demonstrated that the shape recovery ratio reached up to 90% within 10 s after immersing in water at 37 °C, Figure 13. The authors argued that the storage modulus of PHP decreased upon immersing the samples in water because of the disruption of hydrogen bonds in the PHP matrix. 

In the reference [99], the authors prepared a nanocomposite paper based on polyethylene glycol (PEG) as the matrix, with sisal cellulose nanofibers (CNF) and citric acid (CA) as a cross-linking agent using an evaporation-induced self-assembly method. They reported that by controlling the CA content, the water swelling resistance of the prepared composite could be controlled. Moreover, they argued that the prepared material had an improved tensile modulus because of the strong covalent bonds that were formed by the esterification reaction of the carboxyl group on CA and the hydroxyl groups on PEG and CNF cellulose, leading to a stability improvement in the molecular chain. The authors showed that the prepared nanocomposite had a shape recovery rate of 90.2% within 11 s after immersing in water at 25 °C, Figure 14. 

#### 2.2.2. Chemical Activation by Solvents

The activation mechanism of SME for water and organic solvents is the same, whereas the difference lies in the need to prepare solvent responsive SMPs that are more adaptable to a different environment. For example, Tong et al. [100] prepared solvent response gels using copolymerization of hydrophobic butyl methacrylate (BMA) and hydrophilic methacrylic acid (MAA) with cross-linker N,N′-methylenebis(acrylamide) (BIS) with good mechanical properties (tensile strength of 9 MPa and Young’s modulus of 300 MPa). They claimed that the temporary shape of the prepared composite, which was fixed in H_2_O, could be recovered in dimethyl sulfoxide (DMSO) for at least 10 times; and that the actuation rate was 30 s by using solvent exchange, Figure 15.

However, the activation process of SMPs using solvents has some disadvantages, such as the loss in the modulus because of the softening process of the solvent, which leads to a reduction in the mechanical properties. Moreover, the response rate of this type of SMP (recovery time) is considered slow, therefore it is considered the main challenge for using solvent activated SMPs in real application.

### 2.3. Activation by Light 

Light activated shape memory polymers can generate a reversible deformation when exposed to light at any wavelength in the range of 1–10^6^ nm. Since light is considered a source of energy, it is also considered a control signal carrier that can travel for long distances without the need for a transforming medium. Moreover, this type of light activated SMP has great interest due to the advantages associated with many other different light sources, the absence of electromagnetic interference, the possibility of remote control, the ability to be directed to a certain area, and the ability to disperse through optical fibers [101,102,103]. Light activated SMPs can be applied in different applications, such as soft robots, sensors, and biomimetic devices [104,105,106].

The key parameter in this activation type is the presence of light responsive groups (such as cinnamic groups), which under exposure of certain UV wavelengths, can efficiently carry out photo-reversible cycloaddition reactions [12]. Generally, in the programming stage, the coiled macromolecules of the polymer chains in the amorphous phase will be stretched between the netpoints. In the case of light activated SMPs, under exposure to a certain UV wavelength (λ_A_), Figure 16, new chemical netpoints will be formed fixing the temporary shape of the polymer. Then, under the exposure of another UV wavelength (λ_B_), the formed netpoints in the programming stage will be released, which will lead to restoration of the initial shape of the polymer [107]. Since the main process in the programming stage for the thermal activated SMPs is the freezing of the stretched macromolecules in the polymer amorphous phase, for the light activated SMPs, the fixation of the polymer temporary shape occurs by the formation of chemical netpoints because of the light irradiation [12]. 

In the reference [108], the authors demonstrated that the response to light was caused by a change in the conformations (arrangement) of individual groups of macromolecules inside hydrogels, which led to a change in the volume of the entire material. The authors asserted that the reversible change in conformations was carried out on the trans-azo groups upon irradiation with ultraviolet light at a wavelength of 365 nm, which was accompanied by an increase in the volume of the material, Figure 17. Irradiation of the hydrogel with visible light at a wavelength of 430 nm led to a change in conformation at the cis-azo group and reverse compression of the material, Figure 17.

Another type of light activated SMP is the addition of photo-thermal fillers into the matrix of the thermally induced SMPs [101,103,104,109]. The phenomenon that generates thermal energy from electromagnetic radiation is called the photo-thermal effect. 

In the reference [104], a composite that had a photo-thermal effect near infrared (NIR) light was prepared based on poly [ethylene-ran-(vinyl acetate)] (EVA) using rare earth organic complexes of Yb (TTA)3Phen and Nd (TTA)3Phen as photo-thermal fillers. The authors showed that at a relatively short exposure time of the NIR light irradiation, the deformation temperature of EVA, enabled decreased polymer shape recovery, Figure 18. 

In the reference [101], woven carbon fiber was used as a photo-thermal filler for the shape memory epoxy composite and as a reinforcing filler. The authors demonstrated that the shape fixity and recovery ratios of the prepared composite were 100% and 86%, respectively. The authors claimed that by increasing the power density of the NIR radiation, the temperature of the prepared composite increased leading to a faster recovery rate in comparison with the thermal activation method, Figure 19.

It should be noted that applying light activated SMPs is more complicated than photo-thermal activated SMPCs. This is related to the potential properties of the added fillers, which provide the ability to enhance the structural performance, mechanical properties, and the rate and efficiency of SME.

## 3. Applications of SMPs

Due to the attractive properties of SMPs, such as lightweight, low cost, manufacturability, biodegradability, biocompatibility, good mechanical properties, they are attractive materials for use in a wide range of applications as smart materials.

### 3.1. SMPs as Artificial Muscles 

Coiled SMPs that can be activated by different stimuli are considered an attractive material to prepare artificial muscles. These coiled SMPs can be used in a wide range of applications, such as bioinspired actuators, smart hair styles, smart textiles, smart grippers, and crawling soft robotics and sensors. For example, coiled artificial muscles based on oriented ultra-high molecular weight polyethylene (UHMWPE) fibers were prepared in the work [42]. The authors demonstrated that these actuators showed a tensile stroke up to 87% with excellent mechanical properties, Figure 20.

In the reference [43], coiled pH-responsive artificial muscles were based on aligned polyacrylonitrile (PAN) fibers with a contractive actuation stroke of 47%, Figure 21. The authors showed that the prepared artificial muscles had a pH- response due to the introduction of the carboxyl groups to PAN fibers using a hydrolysis method after thermal stabilization. 

### 3.2. SMPs in Aerospace Engineering

To apply any material in aerospace application, a series of parameters should be taken into consideration, such as UV radiation, space debris, surrounding environment, including the presence of ions and electrons, temperature, pressure, and atomic composition, etc. This means that the material can work under the presented hard conditions in aerospace to reduce material damage and increase the system’s reliability. SMPs and their composites have attracted more attention in the last two decades for use in aerospace applications in comparison with traditional metallic and ceramic composites due to their attractive properties [29,30,31,32]. 

SMPs can be applied in aerospace as hinges, booms, wrinkled/slack control components, solar arrays, and deployable trusses. For example, Liu et al. developed a tip-loaded deployable truss (TLD truss) with a gusseted base and auxiliary support frame to carry a heavy tip load based on carbon fabric reinforced cyanate-based shape memory polymer composite as the driving source to unfold the packaged truss, Figure 22. The authors showed that the prepared TLD truss with a tip load of 1.3 kg had the highest stiffness and held the most environmental mechanical stresses [110].

In the reference [111], an integrative hinge based on carbon fiber reinforced shape memory epoxy composites was investigated. The authors argued that the self-deployable tube need about 30 s to recover its original shape for the first hinge (45° angle deformation), whereas for the next three hinges (from 180° to 0) it needed about 60 s with 100% shape recovery ratio, Figure 23.

### 3.3. SMPs in Textile Engineering

Incorporating SMPs into the fabric to prepare smart textile can provide smart features, such as perfect aesthetic appeal, soft textile, comfort, wound monitoring, wetting properties [37,38,39,40,41]. Zhi et al. prepared a shape memory super-capacitor (SMSC), which consisted of shape memory NiTi wires that acted as the current collector, and MnO_2_ and PPy that acted as the electrochemical active materials, Figure 24 [112]. The authors claimed that the prepared SMSC could be thermally activated depending on the human body temperature (around 35 °C) and could restore plastic deformations to the undistorted state within a few seconds.

In the reference [113], the authors developed a reversibly actuatable liquid crystal elastomer (LCE) fibers by using direct ink write (DIW) printing, which could be thermally activated. The authors showed that the prepared LCE fibers had a modulus of 2 MPa and an actuation strain of 51%, Figure 25.

## 4. Conclusion and Prospects 

Being regarded as smart materials, shape memory polymers are the best solution for a wide range of applications, such as artificial muscles, bioinspired actuators, smart clothes, solar arrays, and deployable trusses due to their outstanding properties, such as structural versatility, lightweight, low cost, easy processing, mechanical, biocompatibility, and biodegradability. In this article, the activation of shape memory effect in SMPs by direct and indirect heating, light (photo- and photo-thermal effect), and chemicals (solvents including water) was simply demonstrated, and the advantages and disadvantages of each activation mechanism were reviewed, Table 1. It worth mentioning that these stimuli can be applied for different purposes, for instance, water can be used not only as a solvent but also as a medium for heat transfer, and light can be applied as a cross-linking agent for bond re-establishment and as a photothermal heating agent. 

Since SME activation by heating is still the basic mechanism, thermal conductivity of SMPs will remain the main challenge in finding ways to improve both the thermal and electrical conductivity of the SMPs in the few next years. Therefore, adding different types of fillers to the polymer matrix can lead to the enhancement of its thermal and electrical conductivity. However, the other effects of the added fillers on the polymer properties, especially the structural and mechanical ones, will become the prime direction for future investigations. One of the most important keys in SMP science is the method of preparing the SMPs, such as extrusion, injection molding, chemical cross-linking, and many others. As a result, several promising methods are those that use 3D printing, especially for the preparation of hydrogels. Furthermore, preparation methods affect the polymer structure, the mechanical properties, the durability, etc., which requires attention to the importance of understanding the preparation methods and their influence on SMPs in preparing successful products that are based on SMPs and have the possibility to be applied in real applications.

Currently, most researchers are focusing on developing the activation mechanisms of the one-way SMPs in order to fully understand them. However, two-way reversible SMPs are also considered the main challenge for future studies concerning the recovery time for most SMPs, which requires specific attention to be paid to reduce this parameter. Therefore, understanding the working mechanism, polymer structure, and the effect of added fillers on the activation process of SMPs are the main keys required in the development of applicable SMPs.

## Figures and Tables

**Figure 1 polymers-14-03511-f001:**
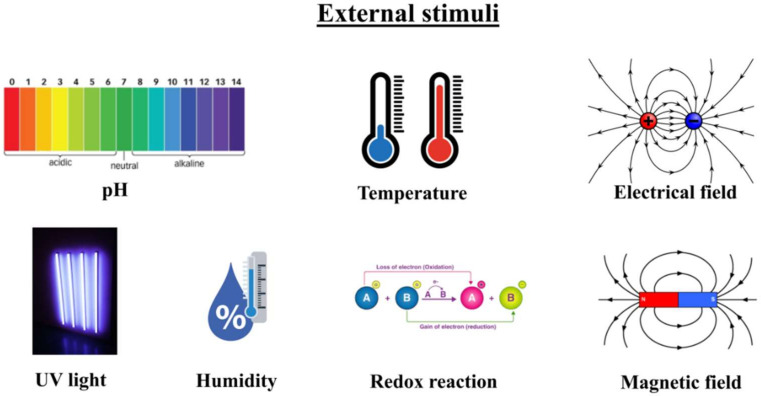
External stimuli for activation of shape memory polymers [8].

**Figure 2 polymers-14-03511-f002:**
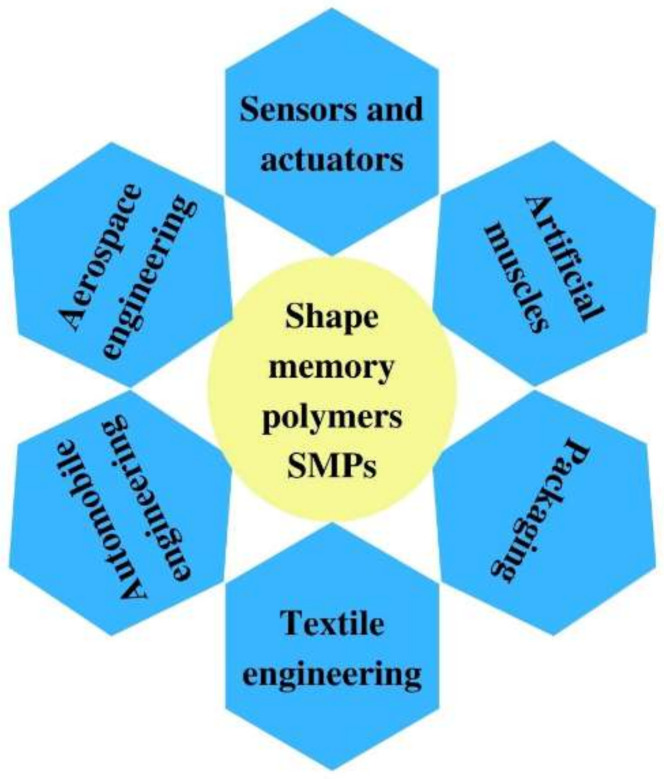
Applications of SMPs.

**Figure 3 polymers-14-03511-f003:**
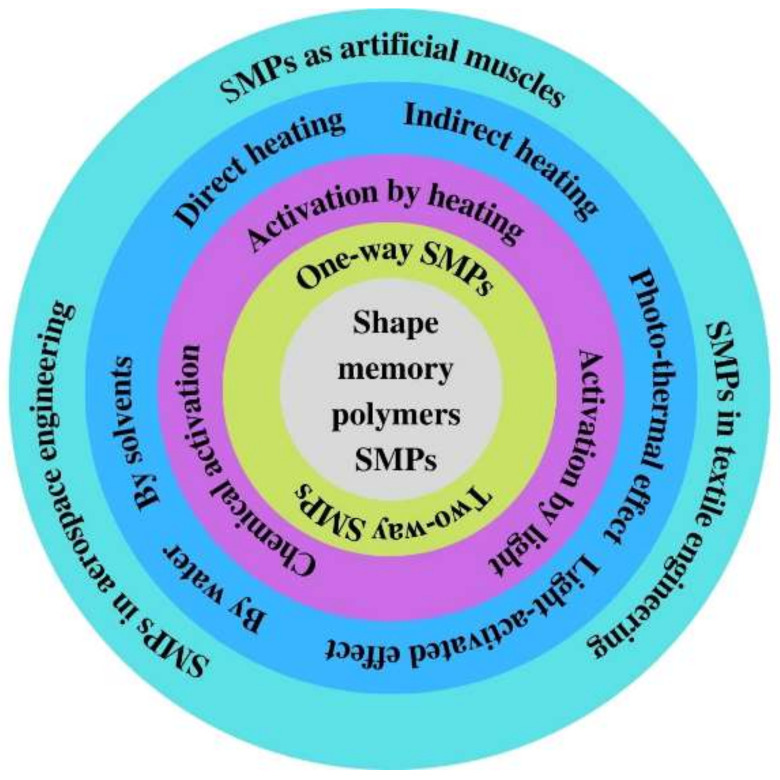
Summary of activation mechanisms of SMPs and their applications.

**Figure 4 polymers-14-03511-f004:**
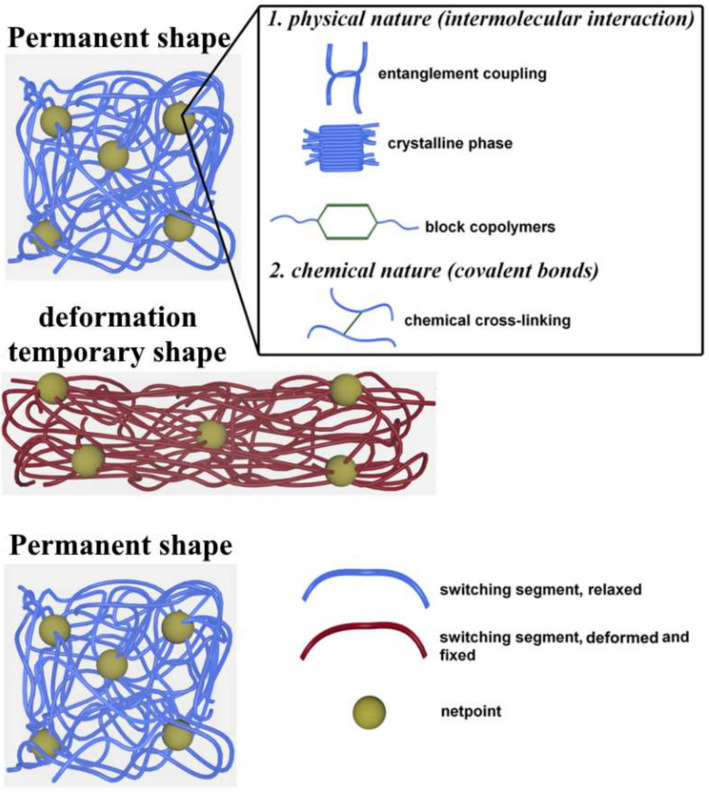
Molecular mechanism for activation of shape memory effect in SMPs.

**Figure 5 polymers-14-03511-f005:**
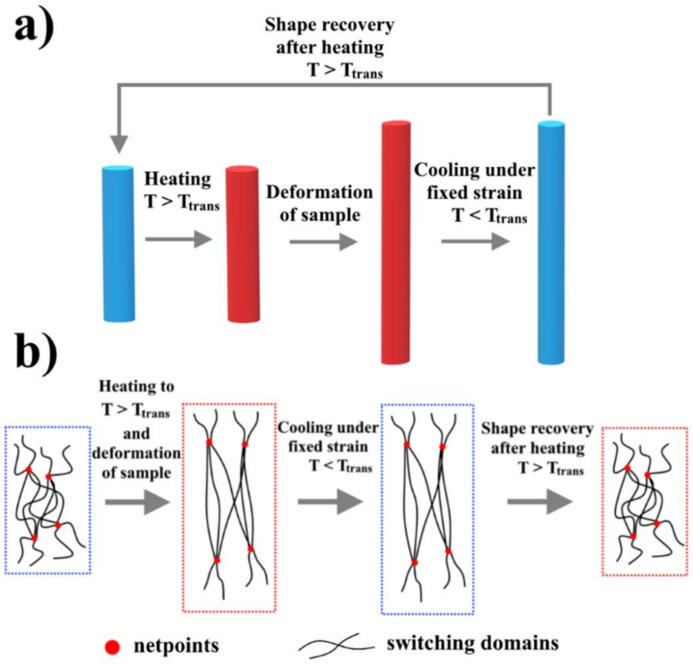
One-way shape memory effect in SMPs at macroscopic level (**a**) and at structure level (**b**).

**Figure 6 polymers-14-03511-f006:**
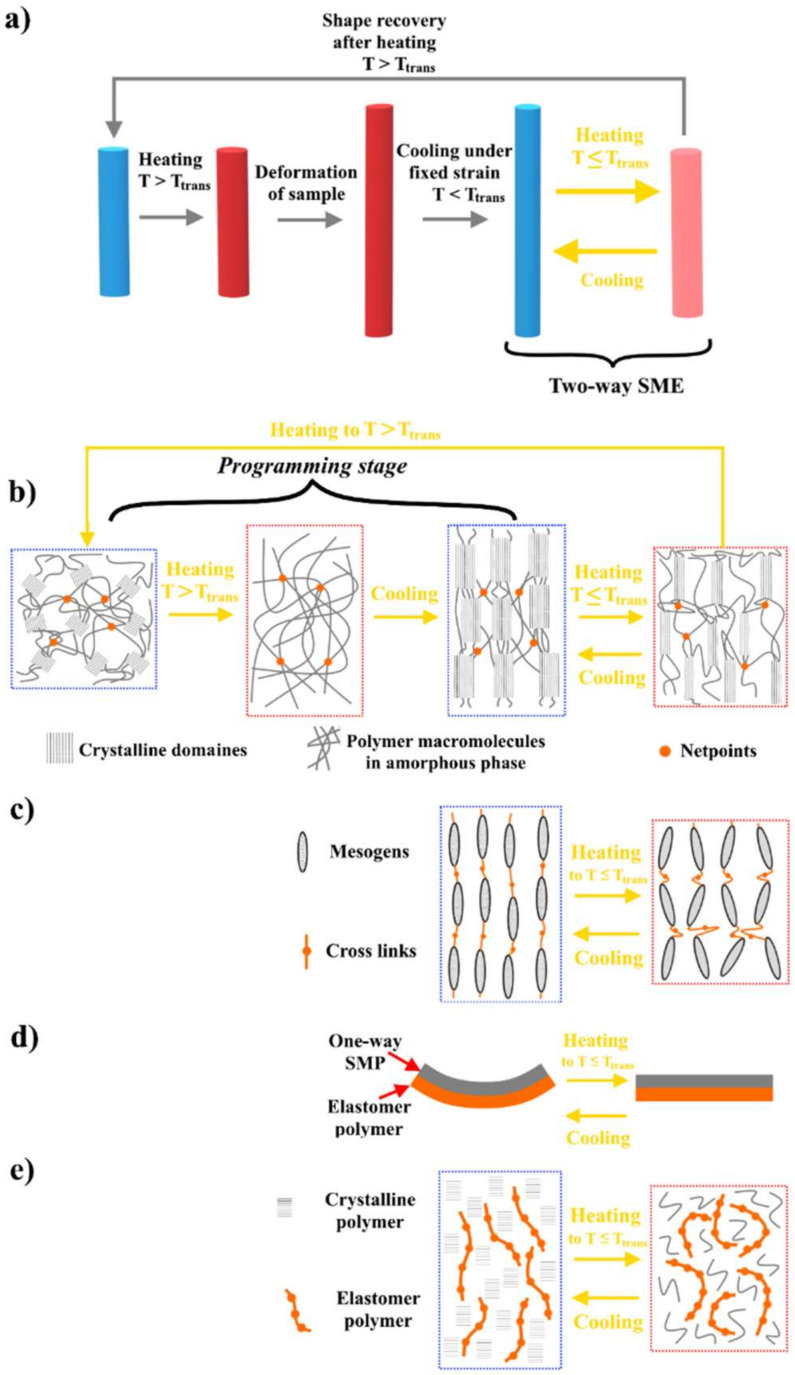
Two-way shape memory effect in SMPs at macroscopic level (**a**) and at structure level for: (**b**) semi-crystalline polymers, (**c**) liquid crystalline elastomers (LCEs), (**d**) multi-layered polymer composites, (**e**) interpenetrating polymers.

**Figure 7 polymers-14-03511-f007:**
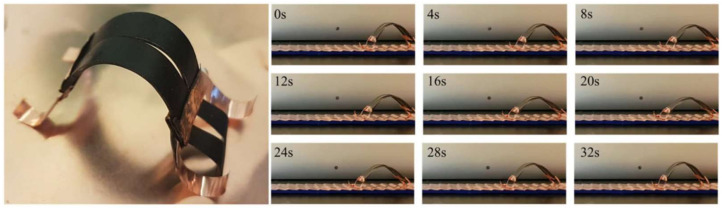
Electrothermal actuator crawling soft robot and actuation behavior [84].

**Figure 8 polymers-14-03511-f008:**
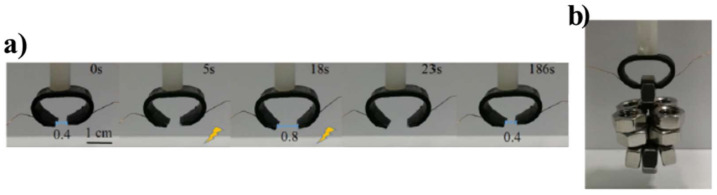
Mechanical gripper based on prepared POE/S-CNT composite: (**a**) Digital pictures of this gripper made with 2 vol % of CNT under a voltage of 36 V dc (on and off), (**b**) Digital picture of the gripper for grabbing nuts. Reprinted/adapted with permission from Ref. [86].

**Figure 9 polymers-14-03511-f009:**
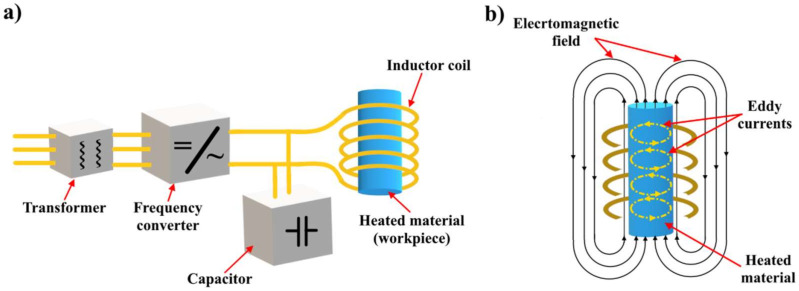
(**a**) Schematic diagram of the inductor, (**b**) The occurrence of eddy currents in an electrically conductive material (SMPCs) under the influence of a high frequency electromagnetic field.

**Figure 10 polymers-14-03511-f010:**
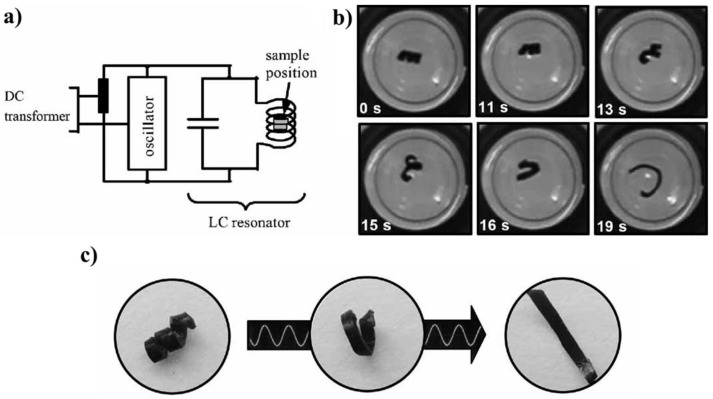
(**a**) schematic representation of the inductor and sample, (**b**) and (**c**) photographs demonstrating the change in the shape of the shape memory polymers under the influence of induction heating. Reprinted/adapted with permission from Ref. [16].

**Figure 11 polymers-14-03511-f011:**
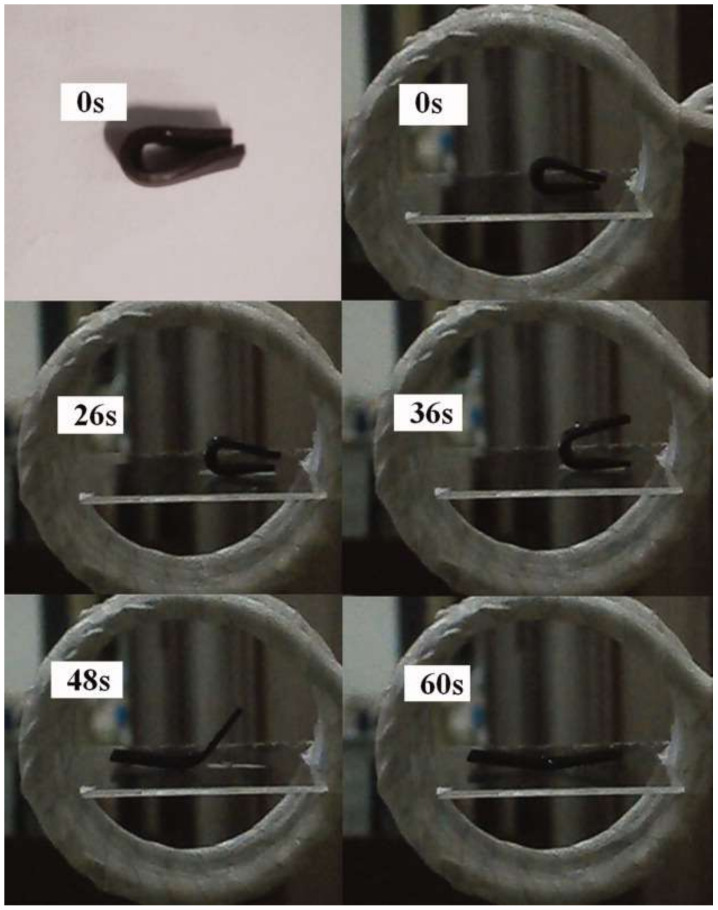
A typical process of shape memory recovery of nanocomposites at 40 °C with alternating magnetic field of 45 kHz. Reprinted/adapted with permission from Ref. [90].

**Figure 12 polymers-14-03511-f012:**
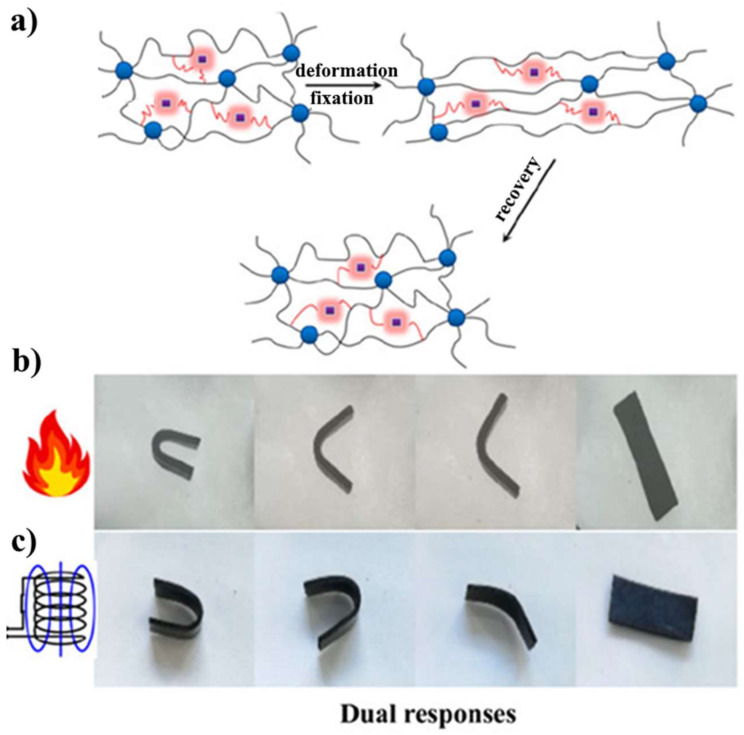
Dual response SME of carboxylic styrene butadiene rubber (XSBR)/ferriferrous oxide (Fe_3_O_4_)/zinc dimethacrylate (ZDMA) composites. (**a**) schematic of activation mechanism of SME, (**b**) activation of SME by direct heating, (**c**) activation of SME by thermal inductive heating. Reprinted/adapted with permission from Ref. [91].

**Figure 13 polymers-14-03511-f013:**
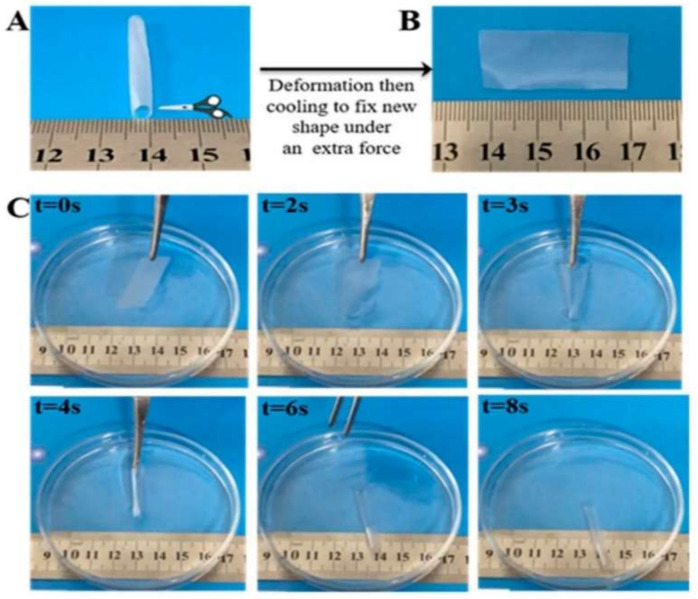
Water responsive shape memory PHA-based polyurethane. (**A**) Original shape of the material, (**B**) deformation process of the material, (**C**) shape memory behavior of the material in water [98].

**Figure 14 polymers-14-03511-f014:**
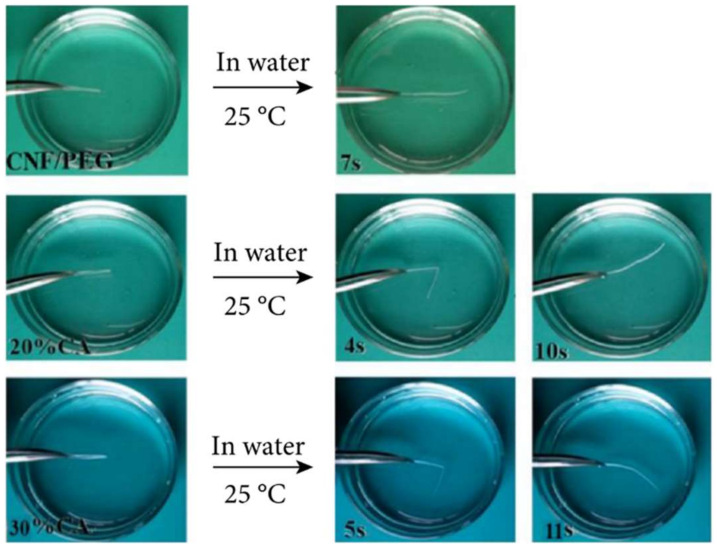
The shape recovery process of the CNF/PEG/CA paper in water [99].

**Figure 15 polymers-14-03511-f015:**
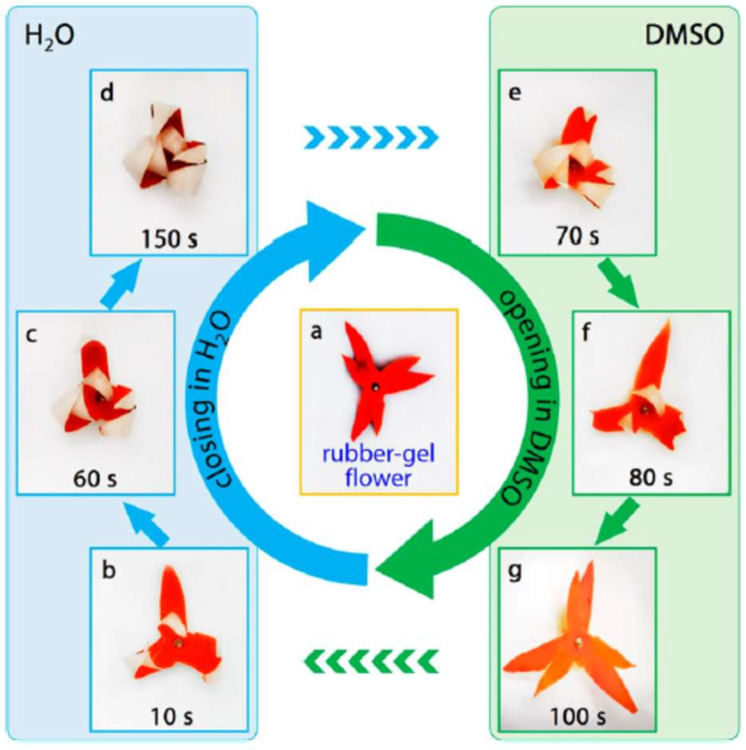
Solvent response of BMA/MAA/BIS gels in water/ DMSO. Reprinted/adapted with permission from Ref. [100].

**Figure 16 polymers-14-03511-f016:**
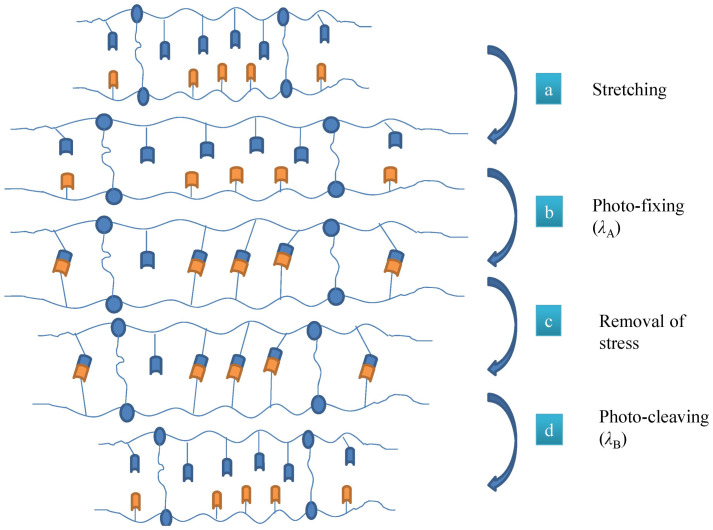
Molecular mechanism of shape-memory effect for light activated SMPs [107].

**Figure 17 polymers-14-03511-f017:**
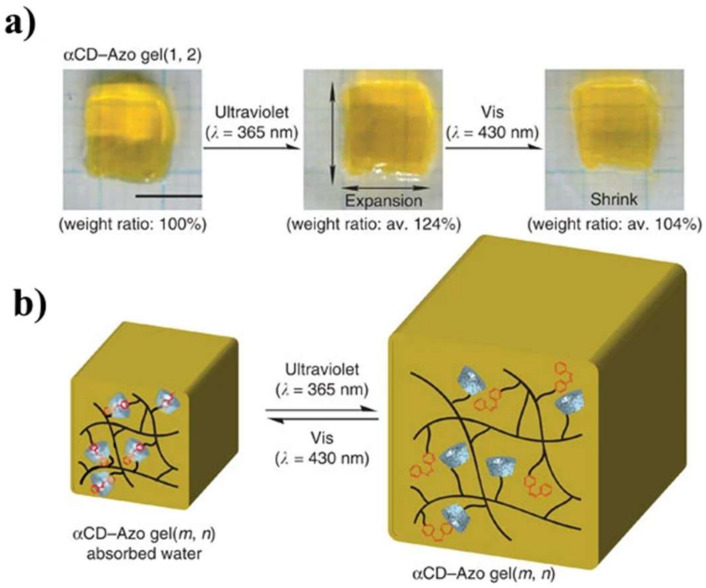
(**a**) Photographs of the experimental sample and (**b**) a schematic representation of the change in the structure inside the hydrogel when irradiated with light of different wavelengths [108].

**Figure 18 polymers-14-03511-f018:**
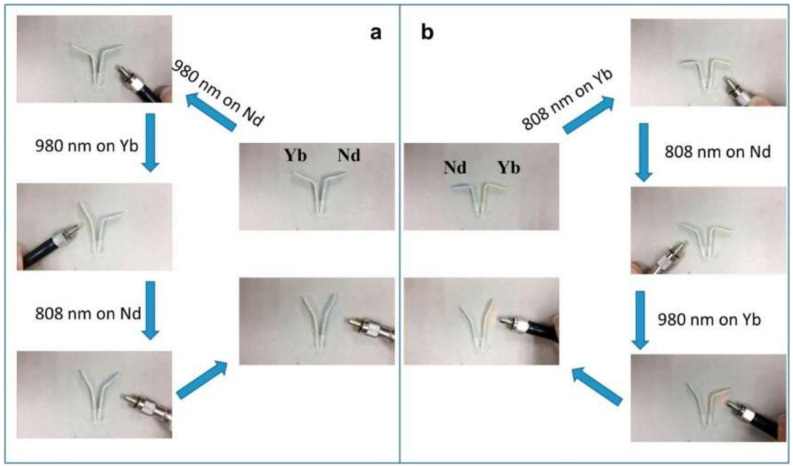
Photographs of selective shape recovery demonstration because of the addition of photo-thermal fillers. (**a**) Irradiation using NIR light of 980 nm first, then by 808 nm, (**b**) irradiation using NIR light of 808 nm first, then by 980 nm. Reprinted/adapted with permission from Ref. [104].

**Figure 19 polymers-14-03511-f019:**
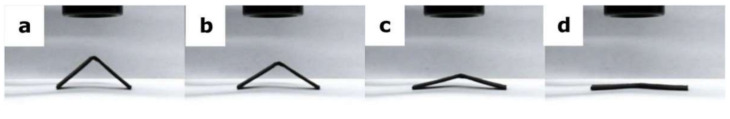
Light activated shape recovery steps for SMPs using a wavelength of 808 nm, (**a**) initial stage with ∼90° bend angle, (**b**) intermediate stage one with ∼115° recovery angle, (**c**) intermediate stage two ∼150° recovery angle, (**d**) final stage with ∼173° recovery angle. Reprinted/adapted with permission from Ref. [101].

**Figure 20 polymers-14-03511-f020:**
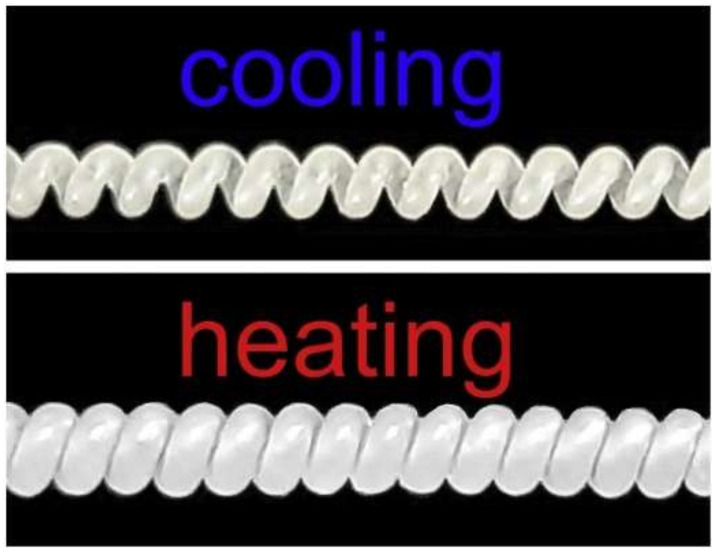
Thermal activated coiled artificial muscles based on oriented UHMWPE fibers. Reprinted/adapted with permission from Ref. [42].

**Figure 21 polymers-14-03511-f021:**
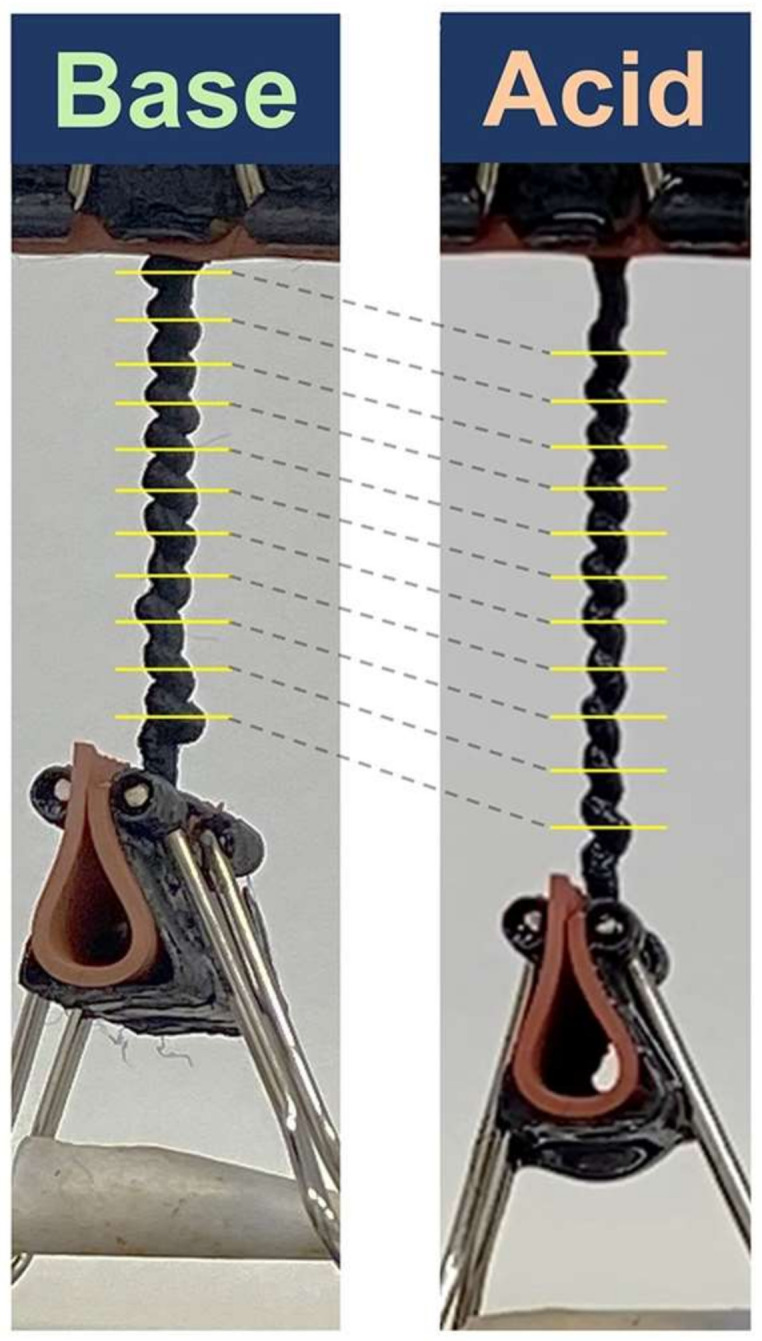
Coiled pH-responsive artificial muscles based on aligned polyacrylonitrile (PAN) fibers. Reprinted/adapted with permission from Ref. [43].

**Figure 22 polymers-14-03511-f022:**
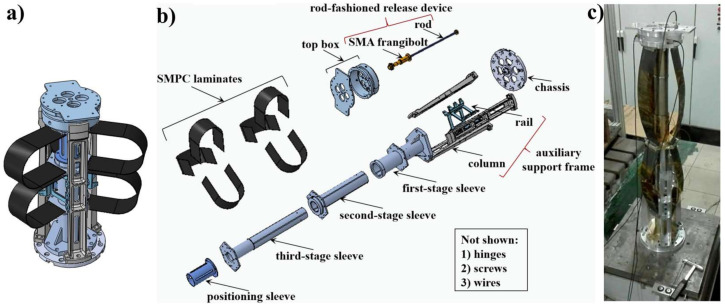
The schematics of the TLD truss (**a**), components of the TLD truss (**b**), real photograph of deployed truss (**c**). Reprinted/adapted with permission from Ref. [110].

**Figure 23 polymers-14-03511-f023:**
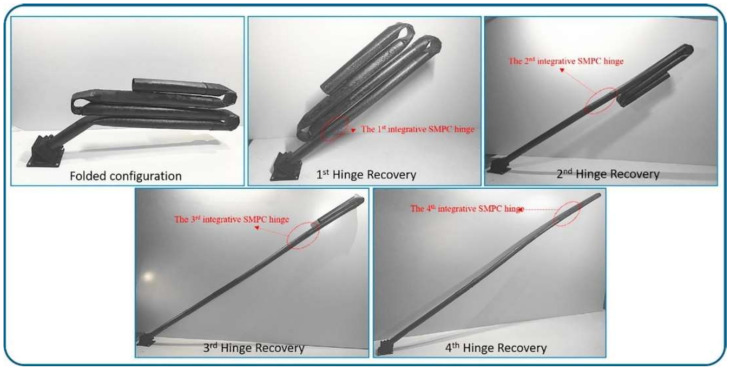
Shape memory recovery process of self-deployable structure. Reprinted/adapted with permission from Ref. [111].

**Figure 24 polymers-14-03511-f024:**
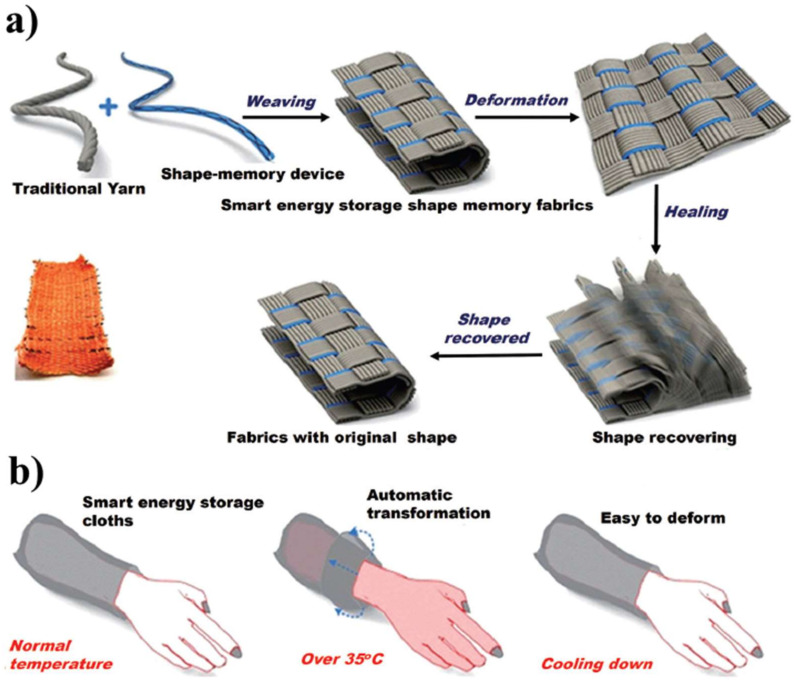
Schematic demonstration of prepared shape memory super-capacitor (**a**), schematic demonstration of this smart shape-memory textile used in a smart cloth (**b**). Reprinted/adapted with permission from Ref. [112]. 2016, Royal Society of Chemistry.

**Figure 25 polymers-14-03511-f025:**
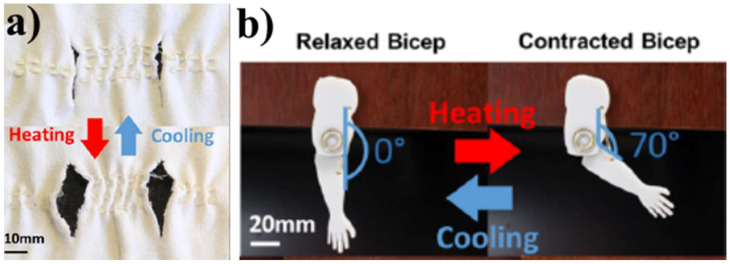
Formation of pores by heating in the modified cotton shirt by LCE fibers (**a**), activation of a single relaxed and contracted bicep muscle fiber using an LCE fiber by heating (**b**). Reprinted/adapted with permission from Ref. [113].

**Table 1 polymers-14-03511-t001:** Activation methods of SMPs and their advantages and disadvantages.

Activation Methods		Activation Mechanism	Advantages	Disadvantages
**By heating**	Direct heating	Heating the polymer higher than its T_trans_	Simple processing. Cheap.	Low thermal conductivity. Slow response.
	Indirect	Activation of SMPs via Joule heating by applying an electrical voltage	Faster actuation response due to enhanced thermal and electrical conductivity. High heating efficiency (>90%).	Filler aggregation leads to uneven dispersion of filler, which in its turn leads to losing the percolated network inside the polymer matrix.
	Induction heating	Activation of the SMPs via eddy currents induced using an alternating electromagnetic field	Faster actuation response due to enhanced thermal conductivity. Non-contact heating. Ability to control the heating rate and heating temperature by changing the frequency of electromagnetic field.	Generation of a strong electromagnetic field, which can interfere with nearby electrical equipment.
**By solvents**	Water	Solvent behaves as a plasticizer, leading to enhancing the chain mobility of polymer, which in its turn leads to a reduction in polymer relaxation time and its glass transition temperature	Simple processing. Bistable.	Reduction in mechanical properties because of softening process of solvent. Slow response.
	Solvent		Solvents can be more adaptable to different environments.	
**By light**	Photo-reversible cycloaddition reactions	Under exposure of certain light wavelengths, SMPs can efficiently carry out photo-reversible cycloaddition reactions due to presence of light responsive groups	Fast response.	Complicated. Limited processing conditions.
	Photo-thermal effect	Generating thermal energy from electromagnetic radiation by adding photo-thermal fillers into SMPs matrix	Fast response. Enhanced structural performance. Improved mechanical properties. Improved efficiency of SME.	Complicated. Filler aggregation.

## Data Availability

No additional data available.

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
