# Peer review of "Shape Memory Polymers as Smart Materials: A Review"

_polymers, 2022, doi:10.3390/polym14173511_

Round 1

Reviewer 1 Report

Shape memory polymers are one of the most famous smart materials with controllable shape changes, it has attracted great attention of researchers and has been applied in many fields. In this review, the three activation mechanisms and applications of shape memory polymers are summarized, and at the end of the article, the author also briefly puts forward his views on the future development of this field. In general, the manuscript is well-organized. However, there are still some issues to be addressed. A moderate revision is suggested before its acceptance.

1.       The introduction of manuscript is too simple. It is suggested that the concept of shape memory polymer be further introduced in the introduction. In addition, one more paragraph to generally introduce the content of this review is better to be provided.

2.       There are too many keywords. It is suggested to delete the words that have poor relevance to the theme of the manuscript.

3.       In introduction section, some recent and highly relevant review articles should be carefully read, introduced and discussed: Recent advances in dynamic covalent bond-based shape memory polymers; Progress in the Field of Water‐and/or Temperature‐Triggered Polymer Actuators; 3D printing hydrogels for actuators: A review; etc.

4.       In figure 1, authors summarized many different stimuli. However, at the end of first paragraph, authors did not list all the stimuli in figure 1. Please further clarify.

5.       It is suggested to add a general pie chart to show the contents of the manuscript more intuitively to the reader at the end of introduction section.

6.       Two tables can be added to summarize activation mechanisms and applications respectively to enrich the contents of the manuscript.

7.       One more section of application of SMPs in actuators should be added with supporting articles: Nanomaterials 12 (1), 53, 2022; In-situ polymerization for mechanical strong composite actuators based on anisotropic wood and thermoresponsive polymer; An Electrospinning Anisotropic Hydrogel with Remotely-Controlled Photo-Responsive Deformation and Long-Range Navigation for Synergist Actuation; A Bamboo/PNIPAM Composite Hydrogel Assembly for Both Programmable and Remotely-Controlled Light-Responsive Biomimetic Actuations; Composite Polymeric Membranes with Directionally Embedded Fibers for Controlled Dual Actuation;  

8.       The conclusion has high similarity with abstract and does not summarize the previous work very well, which should be improved.

9.       There are too few prospects for the future development of shape memory polymers, and more personal opinions for the challenges and solutions should be put forward.

10.   Please carefully check the whole manuscript. There are still some spelling and grammar problems in the manuscript.

11.   Regarding the application of SMPs in textile engineering, one review article should be included: A review of smart electrospun fibers toward textiles.

12.   Authors should carefully recheck the format of references to make sure full information is provided, especially the volume and page numbers.                                                             

Author Response

  1. The introduction of manuscript is too simple. It is suggested that the concept of shape memory polymer be further introduced in the introduction. In addition, one more paragraph to generally introduce the content of this review is better to be provided.

Thank you for your kind comment. Please, check the reformulated and updated introduction:

“Shape memory polymers (SMPs) are smart materials that can be deformed and fixed into a temporary shape and can recover their permanent shape after the release of the external stimulus [1, 2]. Because of their interesting properties, such as excellent structural versatility, lightweight, low cost, easy processing, high elastic strain (more than 200%), biocompatibility, and biodegradability, they attract a high attention in the industrial, aerospace, textile, and medical fields [3 –7]. A lot of external stimuli can be applied on the SMPs (Figure 1), such as heat [9 – 11], light [12, 13], electricity [14 – 17], magnetic fields [18, 19], chemical stimulus (pH changes) [20, 21], humidity [22 – 24], etc.

Recently, many reviews about SMPs have discussed the aspects of SMPs, such as activation mechanisms [9, 25], required molecular structure and changes through the activation of the shape memory effect [26 – 28], their applications, such as aerospace engineering [29 – 32], sensors and actuators [33 – 36], textile engineering [37 – 41], artificial muscles [42, 43], packaging [44, 45], etc., Figure 2. However, a simple and understandable clarification of the structural concepts and activation mechanisms is still missing.

Figure 1. External stimuli for activation of shape memory polymers [8].

Figure 2. Applications of SMPs.

Since SMPs can retain more than two of their temporary shapes during the shape memory effect (SME) process (two or more responses), they are considered convenient candidates for smart systems, which control different conditions and require multiple external stimuli [46]. However, there are some limitations, which obstruct the application of SMPs, such as low thermal and electrical conductivity, weak mechanical properties, and inertness to electromagnetic stimuli in comparison to well-researched shape memory ceramics and metallic alloys [47]. In the last two decades, comprehensive investigations have aimed at developing new polymers with improved SME for better performance [48 – 51]. In other words, modification of polymer structure using different processing methods and adding different fillers to improve the SMPs properties, such as mechanical properties, electrical and thermal conductivity, etc., are considered the main direction of investigations and research all over the world to develop shape memory polymer composites (SMPCs).

Currently, the activation of SME in SMPs can essentially be carried out using three basic types of external stimuli, which are heat (thermo-responsive SMPs); chemicals (chemo-responsive SMPs); light (photo-responsive and photo-thermal responsive SMPs). Thus, this review will focus on the simple explanation of the main principles of SME activation by using the most familiar external stimuli and their latest applications in medical, aerospace and textile applications, Figure 3.

Figure 3. Summary of activation mechanisms of SMPs and their applications.”

  1. There are too many keywords. It is suggested to delete the words that have poor relevance to the theme of the manuscript.

Thank you for your kind suggestion. Please, check the updated keywords, which were also alphabetically reorganized.

Keywords: activation mechanism; amorphous polymer; artificial muscles; electrical conductivity; one-way shape memory effect; semi-crystalline polymer; shape memory polymer; smart material; thermal conductivity; two-way shape memory.”

  1. In introduction section, some recent and highly relevant review articles should be carefully read, introduced and discussed: Recent advances in dynamic covalent bond-based shape memory polymers; Progress in the Field of Water‐and/or Temperature‐Triggered Polymer Actuators; 3D printing hydrogels for actuators: A review; etc.

Thank you for your suggestion. Kindly, be informed that they were added as references.

  1. In figure 1, authors summarized many different stimuli. However, at the end of first paragraph, authors did not list all the stimuli in figure 1. Please further clarify.

Thank you for your comment. Please, be informed that all possible stimuli were summarized in Figure 1, but in the manuscript, only the most important stimuli were discussed. In our point of view, to mention all stimuli will make the manuscript too long and boring for the readers.

  1. It is suggested to add a general pie chart to show the contents of the manuscript more intuitively to the reader at the end of introduction section.

Thank you for your kind suggestion. Please, check the pie chart: Figure 3

  1. Two tables can be added to summarize activation mechanisms and applications respectively to enrich the contents of the manuscript.

Thank you for your suggestion. Please, check the added Table1 and Figure 2:

Table 1. Activation methods of SMPs and their advantages and disadvantages.

Activation methods

Activation mechanism

Advantages

Disadvantages

By heating

Direct heating

Heating the polymer higher than its Ttrans

Simple processing;

Cheap;

Low thermal conductivity;

Slow response;

Indirect

Activation of SMPs via Joule heating by applying an electrical voltage

Faster actuation response due to enhanced thermal and electrical conductivity;

High heating efficiency (> 90%);

Filler aggregation leads to uneven dispersion of filler, which in its turn leads to losing the percolated network inside the polymer matrix;

Induction heating

Activation the SMPs via eddy currents induced using an alternating electromagnetic field

Faster actuation response due to enhanced thermal conductivity;

Non-contact heating;

Ability to control the heating rate and heating temperature by changing the frequency of electromagnetic field

Generation of a strong electromagnetic field, which can interfere with nearby electrical equipment;

By solvents

Water

Solvent behaves as a plasticizer, leading to enhancing the chain mobility of polymer, which in its turn leads to a reduction in polymer relaxation time and its glass transition temperature

Simple processing;

Bistable;

Reduction in mechanical properties because of softening process of solvent;

Slow response;

Solvent

Solvents can be more adaptable to different environments;

By light

Photo-reversible cycloaddition reactions

Under exposure of certain light wavelengths, SMPs can efficiently carry out photo-reversible cycloaddition reactions due to presence of light responsive groups

Fast response;

Complicated;

Limited processing conditions;

Photo-thermal effect

Generating thermal energy from electromagnetic radiation by adding photo-thermal fillers into SMPs matrix

Fast response;

Enhanced structural performance;

Improved mechanical properties;

Improved efficiency of SME

Complicated;

Filler aggregation;

  1. One more section of application of SMPs in actuators should be added with supporting articles: Nanomaterials 12 (1), 53, 2022; In-situ polymerization for mechanical strong composite actuators based on anisotropic wood and thermoresponsive polymer; An Electrospinning Anisotropic Hydrogel with Remotely-Controlled Photo-Responsive Deformation and Long-Range Navigation for Synergist Actuation; A Bamboo/PNIPAM Composite Hydrogel Assembly for Both Programmable and Remotely-Controlled Light-Responsive Biomimetic Actuations; Composite Polymeric Membranes with Directionally Embedded Fibers for Controlled Dual Actuation;

Thank you for your suggestion. Kindly, be informed we previously have reviewed the latest highlights, operating principles, perspectives, and challenges of electroactive materials (EAPs) (actuators) in separate article entitled “Electroactive Polymer-Based Composites for Artificial Muscle-like Actuators: A Review. Nanomaterials 2022, 12, 2272. https://doi.org/10.3390/nano12132272”, since adding a section for SMPs applications in the actuators will make this manuscript more complicated and long.

Please, be informed that some of them were added.

  1. The conclusion has high similarity with abstract and does not summarize the previous work very well, which should be improved.

Thank you for your comment. Please, check the reformulated conclusion:

“Being regarded as smart materials, shape memory polymers are the best solution for a wide range of applications, such as artificial muscles, bioinspired actuators, smart clothes, solar arrays, and deployable trusses due to their outstanding properties, such as structural versatility, lightweight, low cost, easy processing, mechanical, biocompatibility, and biodegradability. In this article, the activation of shape memory effect in SMPs by direct and indirect heating, light (photo- and photo-thermal effect) and chemicals (solvents including water) was simply demonstrated, and the cons and the pros of each activation mechanism were reviewed, Table 1. It worth to mention that these stimuli can be applied for different purposes, for instance, water can be used not only as a solvent but also as a medium for heat transfer, and light can be applied as a cross-linking agent for bond reestablishment and as a photothermal heating agent.

Since the SME activation by heating is still the basic mechanism, thermal conductivity of the SMPs will remain the main challenge, to find ways for improving both the thermal and electrical conductivity of the SMPs in the few next years. Therefore, adding different types of fillers to the polymer matrix can lead to the enhancement of its thermal and electrical conductivity. However, the other effects of the added fillers on the polymer properties, especially the structural and mechanical ones, will become the prime direction for future investigations. One of the most important keys in SMP’s science is the method of preparing the SMPs, such as extrusion, injection molding, chemical cross-linking, and many others, as a result, several promising methods are the ones that use 3D printing, especially for the preparation of hydrogels. Furthermore, preparation methods affect the polymer structure, the mechanical properties, the durability, and so on, that pays the attention to the importance of understanding the preparation methods and their influence on the SMPs to prepare successful products that are based on SMPs and have the possibility to be applied in real applications.

Currently, most researchers focus on developing the activation mechanisms of the one-way SMPs and understanding them. However, two-way reversible SMPs is also considered the main challenge for future studies concerning the recovery time main challenge for most SMPs, which requires specific attention to be paid to reduce this parameter. So, understanding the working mechanism, polymer structure, and the effect of added fillers on the activation process of SMPs are the main keys to the development of applicable SMPs.”

  1. There are too few prospects for the future development of shape memory polymers, and more personal opinions for the challenges and solutions should be put forward.

Thank you for your comment. Kindly, be informed that the required information was added.

  1. Please carefully check the whole manuscript. There are still some spelling and grammar problems in the manuscript.

Thank you for your comment. Kindly, be informed that English was improved and proofread.

  1. Regarding the application of SMPs in textile engineering, one review article should be included: A review of smart electrospun fibers toward textiles.

Please, be informed that it was added.

  1. Authors should carefully recheck the format of references to make sure full information is provided, especially the volume and page numbers.

Thank you for your comment. Kindly, be informed that they were rechecked and corrected.

Reviewer 2 Report

1.      Based on MDPI format, emails from all of the authors are written in black without any underlining.

2.      Please give a “take-home” message as the conclusion of your abstract.

3.      Keywords should be reorganized alphabetically.

4.      Delete table of contents, it is not suitable in format.

5.      Novelty in the current reviews is too weak. The past has seen extensive literature of a lot of written material related to the review of shape memory polymers. It is required to provide more details for more explanation about the present novel in the introductory section.

6.      The last paragraph of the introduction should explain the objective of the present article.

7.      The authors mention UHMWPE without its stands for the abbreviation. Please provide it.

8.      UHMWPE and other materials polymer based have been widely used in the medical implant application, especially on total hip prosthesis. Authors must address this crucial aspect in the introduction and/or discussion section. In addition, to support this explanation, the MDPI-recommended literature should be included as follows: Computational Contact Pressure Prediction of CoCrMo, SS 316L and Ti6Al4V Femoral Head against UHMWPE Acetabular Cup under Gait Cycle. J. Funct. Biomater. 2022, 13, 64. https://doi.org/10.3390/jfb13020064

9.      Please include the limitation of the present review, it is missing.

10.   The reference needs to be enriched from the literature published five years back, literature published by MDPI is strongly encouraged.

11.   In the entire manuscript, the authors occasionally constructed paragraphs with just one or two phrases, which made the explanation difficult to understand. To make their explanation a full paragraph, the authors should expand it. It is advised to use at least three sentences in a paragraph, with the primary sentence coming first and the supporting sentences coming after. See line 236-239 for one of the examples.

12.   English needs to be proofread due to grammatical errors and English style, using the MDPI English editing service would be a solution.

13.   Please ensure that the writers followed the MDPI format correctly, modify the current form, and recheck in addition to any other problems that have been identified.

Author Response

  1. Based on MDPI format, emails from all of the authors are written in black without any underlining.

Thank you for your comments. The correspondence emails were corrected according to the required form. All emails of the authors were added.

  1. Please give a “take-home” message as the conclusion of your abstract.

Thank you for your kind comment. Please, check the reformulated abstract:

“Polymer smart materials are a broad class of polymeric materials that can change their shapes, mechanical responses, light transmissions, controlled releases, and other functional properties under external stimuli. A good comprehension of the aspects controlling various types of shape memory phenomena in shape memory polymers (SMPs), such as polymer structure, stimulus effect and many others, is not only important for the preparation of new SMPs with improved performance, but also useful for the optimization of the current ones to expand their application field. In this era, simple understanding of the activation mechanisms, polymer structure, the effect of the modification of the polymer structure on the activation process using fillers or solvents to develop new reliable SMPs with improved properties, long lifetime, fast response, and the ability to apply them under hard conditions in any environment is considered an urgent topic. Moreover, good comprehension of the activation mechanism of the two-way shape memory effect in SMPs for semi-crystalline polymers and liquid crystalline elastomers is the main key for future investigations. In this article, the principles of the three basic types of external stimuli (heat, chemicals, light) and their key parameters that affect the efficiency of the SMPs are reviewed in addition to several prospective applications.”

  1. Keywords should be reorganized alphabetically.

Keywords were alphabetically reorganized. Please, check the reformulated keywords:

Keywords: activation mechanism; amorphous polymer; artificial muscles; electrical conductivity; one-way shape memory effect; semi-crystalline polymer; shape memory polymer; smart material; thermal conductivity; two-way shape memory.”

  1. Delete table of contents, it is not suitable in format.

Table of contents was deleted.

  1. Novelty in the current reviews is too weak. The past has seen extensive literature of a lot of written material related to the review of shape memory polymers. It is required to provide more details for more explanation about the present novel in the introductory section.

Thank you very much for your kind comment. Please check the reformulated and updated introduction:

“Shape memory polymers (SMPs) are smart materials that can be deformed and fixed into a temporary shape and can recover their permanent shape after the release of the external stimulus [1, 2]. Because of their interesting properties, such as excellent structural versatility, lightweight, low cost, easy processing, high elastic strain (more than 200%), biocompatibility, and biodegradability, they attract a high attention in the industrial, aerospace, textile, and medical fields [3 –7]. A lot of external stimuli can be applied on the SMPs (Figure 1), such as heat [9 – 11], light [12, 13], electricity [14 – 17], magnetic fields [18, 19], chemical stimulus (pH changes) [20, 21], humidity [22 – 24], etc.

Recently, many reviews about SMPs have discussed the aspects of SMPs, such as activation mechanisms [9, 25], required molecular structure and changes through the activation of the shape memory effect [26 – 28], their applications, such as aerospace engineering [29 – 32], sensors and actuators [33 – 36], textile engineering [37 – 41], artificial muscles [42, 43], packaging [44, 45], etc., Figure 2. However, a simple and understandable clarification of the structural concepts and activation mechanisms is still missing.

Figure 1. External stimuli for activation of shape memory polymers [8].

Figure 2. Applications of SMPs.

Since SMPs can retain more than two of their temporary shapes during the shape memory effect (SME) process (two or more responses), they are considered convenient candidates for smart systems, which control different conditions and require multiple external stimuli [46]. However, there are some limitations, which obstruct the application of SMPs, such as low thermal and electrical conductivity, weak mechanical properties, and inertness to electromagnetic stimuli in comparison to well-researched shape memory ceramics and metallic alloys [47]. In the last two decades, comprehensive investigations have aimed at developing new polymers with improved SME for better performance [48 – 51]. In other words, modification of polymer structure using different processing methods and adding different fillers to improve the SMPs properties, such as mechanical properties, electrical and thermal conductivity, etc., are considered the main direction of investigations and research all over the world to develop shape memory polymer composites (SMPCs).

Currently, the activation of SME in SMPs can essentially be carried out using three basic types of external stimuli, which are heat (thermo-responsive SMPs); chemicals (chemo-responsive SMPs); light (photo-responsive and photo-thermal responsive SMPs). Thus, this review will focus on the simple explanation of the main principles of SME activation by using the most familiar external stimuli and their latest applications in medical, aerospace and textile applications, Figure 3.

Figure 3. Summary of activation mechanisms of SMPs and their applications.”

  1. The last paragraph of the introduction should explain the objective of the present article.

Thank you for your suggestion. Please be informed that an explanation was added.

  1. The authors mention UHMWPE without its stands for the abbreviation. Please provide it.

Please, be informed that the full name of UHMWPE was added.

  1. UHMWPE and other materials polymer based have been widely used in the medical implant application, especially on total hip prosthesis. Authors must address this crucial aspect in the introduction and/or discussion section. In addition, to support this explanation, the MDPI-recommended literature should be included as follows: Computational Contact Pressure Prediction of CoCrMo, SS 316L and Ti6Al4V Femoral Head against UHMWPE Acetabular Cup under Gait Cycle. J. Funct. Biomater. 2022, 13, 64. https://doi.org/10.3390/jfb13020064

Thank you for your suggestion. Please, be informed that the suggested reference was added in the introduction section.

  1. Please include the limitation of the present review, it is missing.

Thank you for your comment. Please check the reformulated introduction and conclusion.

  1. The reference needs to be enriched from the literature published five years back, literature published by MDPI is strongly encouraged.

Thank you very much for your suggestion. The references were updated.

  1. In the entire manuscript, the authors occasionally constructed paragraphs with just one or two phrases, which made the explanation difficult to understand. To make their explanation a full paragraph, the authors should expand it. It is advised to use at least three sentences in a paragraph, with the primary sentence coming first and the supporting sentences coming after. See line 236-239 for one of the examples.

Thank you for your comment. The article was reformulated taking your suggestion into. Kindly, check the reformulated paragraph as an example:

“For the amorphous polymers, heating the polymer to its Tg gives the possibility to control its chain mobility, which means the ability to reduce the entropy by fixing its chain network. Whereas heating the amorphous polymer to a temperature higher than its Tg leads to unconstrained shape recovery, which is related to the tendency of the amorphous polymer network to increase the entropy when the crosslinking points return to their original positions after heating [70, 72, 73].”

  1. English needs to be proofread due to grammatical errors and English style, using the MDPI English editing service would be a solution.

Thank you for your comment. English was improved and proofread.

  1. Please ensure that the writers followed the MDPI format correctly, modify the current form, and recheck in addition to any other problems that have been identified.

Thank you for your comment.

Round 2

Reviewer 1 Report

Authors have made a well revision. An acceptance is suggested.

Reviewer 2 Report

The Reviewer recommends this manuscript for acceptance.